# Classes of Lipid Mediators and Their Effects on Vascular Inflammation in Atherosclerosis

**DOI:** 10.3390/ijms24021637

**Published:** 2023-01-13

**Authors:** Valter Lubrano, Rudina Ndreu, Silvana Balzan

**Affiliations:** 1Fondazione CNR-Regione Toscana G Monasterio, 56124 Pisa, Italy; 2Institute of Clinical Physiology, National Research Council, Via G. Moruzzi 1, 56124 Pisa, Italy

**Keywords:** atherosclerosis, polyunsaturated fatty acids, lipid mediators, pro-resolving mediators, ω-3 fatty acid diet

## Abstract

It is commonly believed that the inactivation of inflammation is mainly due to the decay or cessation of inducers. In reality, in connection with the development of atherosclerosis, spontaneous decay of inducers is not observed. It is now known that lipid mediators originating from polyunsaturated fatty acids (PUFAs), which are important constituents of all cell membranes, can act in the inflamed tissue and bring it to resolution. In fact, PUFAs, such as arachidonic acid (AA), eicosapentaenoic acid (EPA), and docosahexaenoic acid (DHA), are precursors to both pro-inflammatory and anti-inflammatory compounds. In this review, we describe the lipid mediators of vascular inflammation and resolution, and their biochemical activity. In addition, we highlight data from the literature that often show a worsening of atherosclerotic disease in subjects deficient in lipid mediators of inflammation resolution, and we also report on the anti-proteasic and anti-thrombotic properties of these same lipid mediators. It should be noted that despite promising data observed in both animal and in vitro studies, contradictory clinical results have been observed for omega-3 PUFAs. Many further studies will be required in order to clarify the observed conflicts, although lifestyle habits such as smoking or other biochemical factors may often influence the normal synthesis of lipid mediators of inflammation resolution.

## 1. Introduction

Atherosclerosis is a lipoprotein-driven, non-resolving chronic inflammatory disease that could cause heart attacks [1,2] and cardiovascular disease [3]. Atherosclerosis begins with the retention of apolipoprotein B–containing lipoproteins within the subendothelial intima of arteries, which leads to the activation of endothelial cells, recruitment of leukocytes, and accumulation of cells, extracellular matrix, and lipids [4].

Macrophages, the major leukocyte type ingesting retained lipoproteins in the subendothelial region, in some cases modified by oxidation [5], become foam cells that help to repair and to clear the tissue from the necrotic cells. However, the prolonged pathogenesis of atherosclerosis is marked by endothelial cell dysfunction, followed by the deposition of lipid-laden macrophages and VSMC proliferation, leading to the occlusion of the arterial lumen. A build-up of fatty deposits inside the arteries (atherosclerosis) and an increased risk of blood clots are usually associated with cardiovascular diseases, a general term for conditions affecting the heart or blood vessels. In fact, in several cardiovascular diseases, a non-resolved acute inflammation [3] could cause chronic fibrotic disease [6,7]. Sometimes, plaque necrosis further increases the inflammatory response [8] and could break down the integrity of the fibrous cap, triggering acute occlusive thrombosis and tissue infarction [8,9]. 

In the past, it was assumed that the reduction or elimination of vascular inflammation was principally caused by the decay or cessation of lipid inducers; therefore, the use of LDL-lowering therapy has been revealed to be very useful. Data from basic and clinical studies indicates that anti-inflammatory treatment could represent an important approach to prevent coronary and cerebral atherosclerotic events. However, despite the availability of this safe therapy, the incidence of cardiovascular disease has been increasing [10]. 

In recent years, the importance of lipid mediators in resolving inflammation has been spotlighted [11,12,13,14,15]. Lipid mediators originate from PUFAs, which represent an important constituent of all the cell membranes in the body. In particular, AA, EPA, and DHA are precursors of several lipid mediators that display pro- or anti-inflammatory properties based on the PUFA, enzyme, and receptor to which they are related. Originally considered only as membrane and energy storage molecules, lipid mediators are now recognized as powerful signaling molecules that regulate a multitude of cellular responses, including growth, apoptosis, and inflammation/infection [16].

However, defects in the resolution of inflammation by lipid mediators have emerged as key causal factors of atherosclerosis [6,17,18]. Specifically, here, we will discuss the role of lipids derived from AA (prostanoids, leukotrienes, epoxyeicosatrienoic acids, and lipoxins), EPA (E-series resolvins), and DHA (D-series resolvins, protectins, and maresins) in relation to cardiovascular inflammation, with the goals of investigating obstacles in the path of vascular inflammation resolution; and identifying novel therapeutic targets in cardiovascular diseases.

Lipid mediators are grouped into three classes based on their structure and function [19].

## 2. Class 1: Eicosanoids

Class 1: Eicosanoids (Prostanoids, Leukotrienes, Epoxyeicosatrienoic acids, and Lipoxins). Eicosanoids derive from membrane phospholipid-released AA which is metabolized by cyclooxygenases (COXs), lipoxygenases (LOXs), and cytochrome P450 (CYP) enzymes to form prostanoids, leukotrienes (LTs), lipoxins (LXs), epoxyeicosatrienoic acids (EETs), and eicosatetraenoic acids (ETEs) [20,21,22,23].

### 2.1. Molecular and Biochemical Characteristic of PROSTANOIDS

The prostanoids comprise the prostaglandins (PGs), the prostacyclins (PGIs), and the thromboxanes (TXA2) (Figure 1).

Prostaglandins show the prefix ‘PG’ followed by a letter A to K, depending on the nature and position of the substituents on the ring, and on the presence or absence of double bonds or hydroxyl groups in various positions in the ring. TXA2 contains an unstable bicyclic oxygenated ring structure [24].

COX-1 is mainly a constitutive enzyme that maintains basal levels of PGs, while COX-2 is an inducible isoform that is regulated by growth factors, cytokines, shear stress, and mitogens. COX-2 is typically upregulated in atherosclerotic lesions [25]. Cyclooxygenases sequentially convert arachidonic acid to the endoperoxides PGG2 and PGH2. PGH2 is further isomerized to each terminal prostanoid by the corresponding synthases. Notably, mPGES (microsomal PGE synthase-1) catalyzes the isomerization of PGH2 into PGE2 biosynthesis. mPGES-1 can couple with both COX-1 and COX-2, and its induced expression increases PGE2, which often correlates with the induction of COX-2 [26]. 

Prostanoids exert a variety of actions through their specific G-protein-coupled receptors, which are differentially expressed in tissues and cells, triggering varied downstream signaling [27]. Among them, the IP, DP, EP2, and EP4 receptors mediate a cAMP rise, whereas EP3 induces a decline in cAMP levels. Finally, FP, and EP1 receptors induce calcium mobilization. However, the concentration or structure of the ligand could change their action [28]. For example, PGI2 stimulates a G-protein-coupled increase in cAMP and protein kinase A, resulting in a decreased intracellular Ca^2+^ concentration [29]. Instead, PGE2, which is produced by a variety of cell types [30], exerts different actions depending on the type of receptor; it increases intracellular cAMP when binding to EP2 and IP receptors, mainly coupled to Gs protein, while it inhibits intracellular cAMP production binding to EP3 receptors, coupled to Gi protein [26]. Finally, its interaction with the EP4 receptor involves PI3K (phosphatidylinositol 3-kinase)/Akt/ERK (extracellular signal-regulated kinase [31]. 

TXA2 is an arachidonic acid metabolite that is a potent vasoconstrictor released by platelet aggregation and involved in atherosclerosis [32,33]. The biological activity of TXA2 is mediated by thromboxane-prostanoid (TP) receptors. TP activation results in the production of inositol 1,4,5-trisphosphate (IP3) and diacylglycerol (DAG) via the activation of phosphatidylinositol (PI)-specific phospholipase C (PI-PLC), through Gq to induce the increase in intracellular Ca^2+^ concentration [33].

### 2.2. Molecular and Biochemical Characteristic of LTs and HETEs

There are six known functional LOX genes in humans (ALOX5, ALOX12, ALOX12B, ALOX15, ALOX15B, ALOXE3), but only three of them, 5-LOX, 12-LOX (platelet-type), and 15-LOX (reticulocyte-type), appear to be important for vascular disorders [34]. 

LTs are a family of eicosanoid inflammatory mediators produced in leukocytes, macrophages and mast cells by the oxidation of AA, and the essential fatty acid EPA by the cytosolic phospholipase A2 and 5-LOX enzymes. 5-LOX, with the aid of the accessory 5-LOX-activating protein (FLAP), catalyzes the conversion of AA to 5-hydroperoxyeicosatetraenoic acid (5-HETE) and then to Leukotriene A4 (LTA4) [35], an unstable intermediate, which can be either metabolized by LTA4 hydrolase to LTB4 (see structure in Figure 1), a potent chemoattractant, or conjugated to glutathione by LTC4 synthase (LTC4S), producing cysteinyl LTs [36]. The LTs exert their actions through interaction with specific 7-transmembrane G-protein-coupled cell surface receptors, BLT1 and BLT2, and the CysLT1 receptor [37]. 

12,15HETEs:12/15-LOX metabolizes arachidonic acid to form 15(S)-hydroperoxyeicosatetraenoic acid (15(S)-HPETE) and 12(S)-hydroperoxyeicosatetraenoic acid (12(S)-HPETE). Both 15(S)-HPETE and 12(S)-HPETE are further reduced by cellular glutathione peroxidase to their corresponding hydroxy analogs: 15-hydroxyicosatetraenoic acid (15(S)-HETE) and 12-hydroxyeicosatetraenoic acid (12(S)-HETE), respectively. Both 15(S)-HPETE and 15(S)-HETE bind to and activate LTB4 receptor 2 and peroxisome proliferator-activated receptor γ (PPAR) [38]. 

### 2.3. Molecular and Biochemical Characteristic of Lxs

Lipoxin A4 (LXA4) and its isomer Lipoxin B4 (LXB4)(see structure in Figure 1) are the principal LXs formed in mammals. They are enzymatically derived from the endogenous ω-6 fatty acid, AA, by 5-LOX and either 12-LOX or 15-LOX enzymes. Additionally, the aspirin-induced acetylation of COX2 can function through a lipoxygenase instead of an endoperoxidase, resulting in the generation of aspirin-triggered lipoxin A4 (15-epi-LXA4) and aspirin-triggered lipoxin B4 (15-epi-LXB4) [39,40]. 

LXA4 and LXB4 are produced principally in human mucosal tissues, in blood vessels, and in platelets [22], and are present in all tissues via transcellular interactions between platelets, neutrophils, and resident cells [41]. LXs were first isolated and identified as inhibitors of polymorphonuclear neutrophil infiltration and as stimulators of the non-phlogistic recruitment of macrophages [42,43]. LXA4 regulates cellular functions through the activation of specific receptors (LXA4 receptor/formyl peptide receptor 2 and G-protein-coupled receptor 32) expressed by neutrophils and monocytes [44].

### 2.4. Molecular and Biochemical Characteristic of EETs

EETs are metabolites of AA by cytochrome CYP epoxygenases [45,46] that show beneficial effects in cardiovascular diseases attributed to anti-inflammation, vasodilation, and natriuresis [47].

### 2.5. Role of Prostanoids, LTs, HETEs, LXs, EETs in Atherosclerotic Progression

Prostanoids: Studies performed in mice through genetically manipulated enzymes and receptors highlighted the many physiological and pathological functions of prostanoids in atherosclerosis (see Table 1).

For example, PGD2 is involved in sleep, in allergies, and adiposity [60]; PGF 2α in childbirth and fibrosis; and PGE2 is involved in many pathological events, such as inflammation, fever, pain, and cancer. The potent vasoconstrictor TxA2 seems to be involved in coronary ischemia.; in contrast, the potent vasodilator PGI2 is involved in antithrombosis [61] and regulates coronary blood flow in humans [62,63] (see Table 2).

In mice, TxA2 promotes, while PGI2 prevents, atherogenesis through the control of platelet activation and leukocyte–EC interaction [48]. At the same time, PGE2 can dilate or constrict large blood vessels, depending on the vessel type and on the receptor subtypes involved (EP2, EP4 or EP1, EP3) and receptor, respectively [77,78,79,80]. mPGES, which catalyzes the isomerization of PGH2 into PGE2, represents a possible target against inflammation and thrombotic risk, because its inhibition augments PGI2 levels. [77]. Studies show that male mice that are deficient in both LDL receptor and mPGES-1 delay atherosclerosis development [49] and reduce abdominal aneurysm events induced by angiotensin II [50,77]. The deletion of mPGES-1 attenuates injury-induced neointima formation in mice by the augmented levels of PGI2, a known restraint of the vascular response to injury, acting via the IP receptor. On the other hand, the global deletion of mPGES-1 exacerbates MI/R injury, despite the increased biosynthesis of PGI2 [81]. Mechanistically, it has been reported that mPGES-1–derived PGE2 and the endothelial EP4 receptor preserve microcirculatory perfusion in MI/R through their vasodilatory effect on arterioles [51]. 

COX-1 and -2 are mediators of pain, fever, and inflammation and the primary targets of nonsteroidal anti-inflammatory drugs (NSAIDs). However, studies in mice using COX-2 knockout or COX-2 derived PGI2 receptor deletion showed elevated blood pressure and thrombogenesis [82]. Moreover, COX-2-selective drugs such as coxibs, rofecoxib, celecoxib, and lumiracoxib NSAIDs, inhibit cyclooxygenase-2 (COX-2), thereby suppressing the biosynthesis of PGI2, and increasing the risk of thrombotic events such as in myocardial infarction and stroke [83,84]. 

LTs: Besides their principal role in asthma and in inflammation, LTs can also increase cardiovascular risk, being important mediators of the inflammatory process in cardiovascular pathologies [64,65]. In particular, cysteinyl-LTs seem to be a potent coronary artery vasoconstrictor that stimulates the proliferation of arterial smooth muscle cells and increases the P-selectin surface expression of endothelial cells [36,64]. Increased levels of LTs are present in patients with myocardial infarction (MI), stroke, atherosclerosis, and aortic aneurysms; and the urinary excretion of LTE4 has been reported to be increased in patients with cardiac ischemia [64,85]. Interestingly, two human genetic studies showed a relationship between polymorphism of the 5-LOX pathway and relative risk for MI, stroke, and atherosclerosis, focusing attention on LTs in the pathogenesis of these diseases [86]. 

HETEs: Increasing evidence highlights the controversial nature of 12/15-LOX in inflammation, as its metabolites possess both pro- and anti-inflammatory properties [87]. The 12/15-LOX metabolite 12(S)-HETE is a potent, pro-inflammatory chemoattractant for neutrophils and leukocytes [38,88]. The 12/15-LOX-derived metabolites 12(S)-HPETE and 15(S)-HPETE were reported to inhibit prostacyclin synthase activity, thereby decreasing the levels of vasodilator prostacyclin [89]. In addition, it was suggested that 12/15-LOX reduces the bioavailability of nitric oxide, thus promoting vasoconstriction [90,91]. At the same time, several reports demonstrate the anti-inflammatory properties of 12/15-LOX and its metabolites [92]. In fact, in human coronary arteries, 12(S)-HETE induced vascular smooth muscle cell hyperpolarization through the activation of large conductance KCa (BKCa) channels, resulting in relaxation [66]. 

Studies of 15-lipoxygenating LOX-isoforms (ALOX15, ALOX15B) indicate that their regulation of vascular tone plays an important role in the development of hypertension [66]. It has been reported that ALOX15-deficient mice exhibited higher resistance towards L-NAME- and high-salt-induced hypertension than wild type controls, although systolic blood pressures did not differ [52]. In support of this, ALOX15-deficient mice treated with wild-type peritoneal macrophages, which are a major source of ALOX15 in mice, lost their resistance toward L-NAME-induced hypertension. Analyzing human atherosclerotic plaques revealed a significant increase in 12/15-LOX expression [93]. It was also reported that ALOX5 and ALOX15B were expressed at high levels in advanced atherosclerotic lesions [94]. However, the role of 12/15-LOX in human atherosclerotic lesion development is still not yet clear. Other studies showed an increased 12/15-LOX expression with decreased lesion severity in human clinical samples. Specifically, a polymorphism in 12/15-LOX promoter (a C to T substitution at position −292), which promotes its increased expression and activity, was found to be associated with a reduction in the risk of atherosclerosis [95].

EETs: EETs are expressed in the heart, especially in the endothelium, and possess beneficial effects against inflammation, fibrosis, and apoptosis, which could combat cardiovascular diseases [96]. Some studies showed that EETs mediate the vasodilation of vascular smooth muscle by activating Ca^2+^-activated K^+^ channels [97], and that its vasodilator effect is independent of nitric oxide (NO) in response to bradykinin [98]. In animal models of hypertension, EETs synthesis was inhibited [99], while it was upregulated by soluble epoxide hydrolase (sEH) inhibitors or by CYP gene overexpression, which in turn decreased blood pressure [100,101]. Interestingly, in Ang II-induced hypertension, sEH expression was upregulated, while EETs were downregulated [53]. 

Other studies showed that sEH inhibition reduced cholesterol and low-density lipoprotein (LDL) levels, while it increased high-density lipoprotein (HDL) levels [102], preventing atherosclerosis in several models, including ApoE −/− mice and high-fat diet (HFD)-induced models [103]. Lower EET levels were observed in obstructed CAD patients compared with patients with no apparent CAD [67]. Furthermore, some authors provided evidence that treatments with sEH inhibitors could reduce the infarct size and chronic cardiac remodeling post MI [104] and that the elevation of EETs might represent a promising therapeutic strategy combating chronic cardiac remodeling post MI and heart failure [105,106]. Already in 2006, Xu et al. reported that treatment with sEH inhibitors prevented the development of cardiac hypertrophy after 3 weeks of the transverse aortic constriction in mice, and reversed the development of cardiac hypertrophy caused by chronic pressure overload [107]. 

Moreover, Althurwi et al. demonstrated that the elevation of EET levels by the overexpression of YP2J2 prevented the initiation of cardiac hypertrophy through the NF-κB-mediated mechanism [108,109]. In particular, in vitro studies showed that the overexpression of CYP2J2 or the increasing levels of 14,15-EET (see structure in Figure 1) ameliorated oxidative-stress-reducing apoptosis and inflammation [110]. However, despite the beneficial effect of EETs as anti-inflammatory mediators, because of their instability, it was impossible to develop corresponding exogenous drugs; several efforts with unremarkable results have been made to increase the level of endogenous EETs, including gene therapy, and EET analogs. Only sEH inhibitor and CYP2J2 gene upregulation seem to provide promising data. 

LXs: LXA4 and LXB4 exert potent anti-inflammatory and resolution actions [111,112]. We will discuss these, together with the lipid mediators, in Class 3. 

## 3. Class 2: Lysophospholipids

Lysophospholipids (LPLs) are bioactive signaling lipids that are generated from the phospholipase-mediated hydrolyzation of membrane phospholipids (PLs) and sphingolipids (SLs). Lysophosphatidic acid (LPA) and sphingosine-1-phosphate (SP1) are two of the best-characterized LPLs which mediate a variety of cellular physiological responses via specific G-protein-coupled-receptor (GPCR)-mediated signaling pathways. Lysophospholipids include the platelet activating factor (PAF), and the LPA and the SP1 (Figure 2) [113,114,115,116]. 

### 3.1. Molecular and Biochemical Characteristics of PAF, LPA and SP1 

PAF: PAF is 1-o-alkyl-2-acetyl-sn-glycero-3-phosphocholine, synthesized from 1-O-alkyl-lysophosphatidylcholine (LPC) by the phospholipase A2 (PLA 2) enzyme [68]. Acetylation by LPC acyltransferase 2 (LPCAT2) terminates PAF biosynthesis [117]. Inactivation of PAF occurs by plasma and cellular acetylhydrolase, a special group of the PLA 2 family [118]. PAF binds to a unique G-protein-coupled seven-transmembrane receptor [119,120] that is highly expressed in cells within the cardiovascular system and central nervous system [121]. It causes many inflammatory reactions and allergic responses, including increased platelet activation, leukocyte adhesion, chemotaxis, and vascular permeability [122,123]. 

LPA: LPA belongs to the glyceryl-based lysophospholipid family [124] and is produced externally to cells, specifically by LPC or autotaxin (ATX; lysophospholipase D) [125], and is degraded by lipid phosphatases [126].

SP1: SP1 is structurally similar to LPA, with the one difference of having a sphingosine backbone rather than glycerol. Sphingolipids are essential structural constituents of the plasma membrane of eukaryotic organisms and some bacteria, where they are highly enriched along with cholesterol in microdomains or lipid rafts. In response to a wide range of stressful stimuli, membrane sphingomyelin are rapidly metabolized to the bioactive sphingolipid intermediate, ceramide, and subsequently to sphingosine phosphorylation by two sphingosine kinases, resulting in the formation of SP1 [127].

### 3.2. Role of PAF, LPA, and SP1 in Atherosclerotic Progression

PAF: Experiments on animal models showed that PAF plays an important role in the pathogenesis of cardiovascular disorders, and some studies performed in humans reported elevated PAF levels in patients, with increased risk of ischemic events in those with coronary artery disease [68,69,70] (see Table 2). In particular, PAFs seems to stimulate reactive oxygen and nitrogen species, thus triggering the oxidative and nitrosative stress pathways, inducing vasoconstriction [128]. Various substances such as thrombin, angiotensin II, vasopressin, histamine, leukotrienes, and hydrogen peroxide cause the production of PAF in endothelial cells [129]. Other studies suggest that PAF plays a role in the evolution of myocardial injury observed during reperfusion [54] (see Table 1). In conclusion, the effects of PAF are closely associated with atherosclerosis, through various mechanisms such as inflammation, endothelial dysfunction, oxidative and nitrosative stress, and platelet reactivity. 

LPA: LPA exerts several effects on vasculature [130,131] resulting in vasoconstriction and increased blood pressure [132]. This happens through the activation of at least six known G-protein-coupled LPA receptors (LPARs) [133] and also peroxisome proliferator-activated receptor γ (PPARγ) [134], and P2Y10 [135]. Compared to normal arteries, individual LPARs are differentially expressed in human atherosclerotic lesions suggesting a distinct role in the initiation and progression of atherosclerosis [136]. Pathological conditions that involve platelet activation, such as inflammation and atherosclerosis, seem to increase LPA production [137]. LPA is elevated in the serum of patients with acute heart attacks [71] and with acute coronary syndrome [72] and is found accumulated in atherosclerotic plaques [73]. 

In the vasculature, platelets are one of the major sources of LPA production [138], and LPA in turn can contribute to thrombosis, a critical pathogenic condition of vascular diseases. LPA seems also to increase the permeability of the endothelium. In fact in HUVECs LPA via LPAR6-Gα13-RhoA-ROCK pathway induces actin stress fiber formation and increases cell permeability [139,140]. Finally, LPA promotes the progression of atherosclerosis via its various effects on inflammatory cells; for example, promoting inflammatory cell migration [141], cytokine secretion [142], and macrophage transformation [143]. LPA also regulates macrophage-related processes, ox-LDL uptake [143], and matrix metalloproteinase (MMP) secretion [144]. MMPs, particularly MMP9, are principle mediators of extracellular matrix production and degradation, which contribute to atherosclerotic plaque instability [145]. Key proteins in LPA homeostasis are increasingly dysregulated in the plaque during atherogenesis, favoring intracellular LPA production. This might at least partly explain the observed progressive accumulation of this thrombogenic proinflammatory lipid in human and mouse plaques. Thus, intervention in enzymatic LPA production may be an attractive measure to lower intraplaque LPA content, thereby reducing plaque progression and thrombogenicity [55].

SP1: S1P has emerged as a potent signaling molecule that regulates diverse cellular processes including cell proliferation, survival, differentiation, and migration [146], maintaining vascular integrity. S1P stimulates COX 2, suggesting the existence of a functional crosstalk between the sphingolipid and eicosanoid pathways [147]. Moreover, many of the cardioprotective functions of HDL, such as vasodilation, angiogenesis, and the endothelial barrier function, may be attributable to its S1P cargo [148]. In a mouse model of acute myocardial infarction, treatment with an engineered S1P chaperone (ApoM-Fc) attenuated myocardial damage after ischemia/reperfusion injury [56]. 

Additionally, ceramides, that belong to sphingolipids, are grouped in a class of bioactive lipids which are present during vascular inflammation [149], but they have opposing effects on vascular endothelium compared to S1P [150,151]. Ceramide properties mainly involve the regulation of apoptosis, and their signaling is potentially linked to atherosclerosis progression and pathology. Growing evidence indicates that circulating ceramides may predict acute cardiovascular risk, and may potentially be more predictive than LDL cholesterol [152]. 

## 4. Class 3: ω-3 PUFA Derivatives

Anti-inflammatory lipid mediators derive from dietary omega-3 PUFA, specifically the ω-3 fatty acids found in fish oil. They include resolvins (Rvs), protectins(Ps), and maresins (MaRs) (Figure 3) [153,154]. 

### 4.1. Molecular and Biochemical Characteristics of Rvs, PDs and MaRs

Rvs: Rvs are divided into two groups: E-series derived from eicosapentaenoic acid and RvD-series from docosahexaenoic acid [155]. The biosynthesis of RvEs is initiated by vascular endothelial cells that convert EPA into 18R-hydroperoxyeicoapentaenoic acid and 18S-hydroperoxyeicoapentaenoic acid (18-HEPE), and proceeds via a transcellular biosynthesis via acetylated COX-2 and cytochrome P450 [156]. These intermediates are rapidly taken up by human neutrophils and metabolized to RvE1 and RvE2 by 5-lipoxygenase. 8-HEPE can directly be transformed into RvE3 through the action of 12/15-LOX by leukocytes [157]. 

In addition, another EPA–derived E-series resolvin, RvE4, is produced by human neutrophils and macrophages by 5-LOX, and successively by a second enzymatic lipoxygenation, to yield a hydroperoxyl group at the carbon C-5 position [158]. RvE4 activates G-protein-coupled receptors such as Chemokine-like receptor 1 (ChemR23) and antagonizes the BLT1 receptor, which usually mediates the proinflammatory actions of LTB4 [159,160,161]. Upon selective binding to the receptors, RvE1 attenuates nuclear factor-kappaB signaling and the production of pro-inflammatory cytokines, including tumor necrosis factor alpha. RvDs target G-protein-coupled receptor 32, LxA4 receptor/formyl peptide receptor 2 receptors, Formyl-Peptide Receptor 2/3/LXA4 Receptor, and CB2 receptors that are expressed on platelets and polymorphonuclear neutrophils. The activation of CB2 leads to the inhibition of P-selectin expression, thus decreasing polymorphonuclear neutrophil chemotaxis [44]. 

PD: Protectins are biosynthesized via a lipoxygenase-mediated pathway from Docosahexaenoic acid and it is converted into an intermediate that contains 17S-hydroxyperoxide, and successively, by leukocytes, into 10,17 Dihydroxy-docosahexaenoic acid, known as PD1 [162,163] or neuroprotection [164], for its protective role in the nervous and immune systems [165,166]. PD1 is also produced by human peripheral blood lymphocytes with a T-helper 2 phenotype; it reduces tumor necrosis factor alpha and interferon gamma secretion, blocks T-cell migration, and promotes T-cell apoptosis [167]. 

A novel protectin-synthesis pathway, called aspirin-triggered PD1, utilizes aspirin-triggered cyclooxygenase-2 to synthesize epimeric 17 R-hydroxyperoxide from docosahexaenoic acid [168,169]. It has shown a positive interaction with CB2 and peroxisome proliferator-activated receptor family receptors [164,170]. 

MaRs: Maresins form the third-largest family of pro-resolving lipid mediators (PRLMs) derived from DHA20. The biosynthesis of maresins occurs primarily in M2 macrophages and is initiated by a reaction involving human macrophage 12-lipoxygenase (12-LOX), which is a key enzyme in the synthesis of maresins [169,171].

### 4.2. Role of PRLMs in Atherosclerotic Progression 

PRLMs such as RvE and RvD, PD, MaR, and LX induce a biochemical transition from inflammation to resolution, taking the place of prostaglandins and leukotrienes in exudates at the tissue level [8,169,172,173]. RvE1-E4 are more prominently involved in reducing inflammation by increasing the efferocytosis of apoptotic cells by macrophages, blocking leukocyte recruitment into inflamed tissues (Figure 4).

Increased plasma levels of RvE1 have been observed in individuals taking aspirin or eicosapentaenoic acid, resulting in the amelioration of the clinical signs of inflammation [74,75]. D-series have been shown to reduce inflammation by decreasing platelet–leukocyte adhesion [174]. 

Protectins reduce polymorphonuclear neutrophil transmigration through endothelial cells and enhance the clearance (efferocytosis) of apoptotic polymorphonuclear neutrophils by human macrophages [170]. 

Maresins were identified as molecules produced by macrophages with homeostatic functions in resolving inflammation. In fact, the macrophage phagocytosis of apoptotic cells triggers the biosynthesis of maresin-1 [175]. Maresin-1 stimulates efferocytosis with human cells and also has regenerative functions [176]. Finally, LXA4 and B4 exert potent anti-inflammatory and resolution actions [110].

Temporal changes in the expression, activity, and localization of enzymes such as 4- and 15-lipoxygenases (LOXs) appear to activate the “switch” responsible for the resolution actions [173,177,178]. Consequently, PRLMs coordinate crosstalk between leukocytes and local cell populations, promoting an M1–M2 phenotypic transition in macrophages, which is central to tissue repair. PRLMs further promote resolution by positive-feedback effects on LOX activity and their receptor expression in leukocytes [177,178,179,180]. Data from murine and rabbit models showed that pro-resolving lipid mediators decrease atheroprogression and promote plaque stability [18,172,180,181]. 

Alternatively a relative “resolution deficit” may contribute to disease progression or limit clinical responses to existing therapies and would therefore suggest opportunities for monitoring and/or treatments that incorporate lipid mediators [74,179,181,182,183]. In fact, circulating levels of PRLMs, in particular 15-epi-lipoxin A4 (15-epi-LXA 4), were significantly lower in individuals with peripheral arterial disease and correlated inversely with disease severity [74,76](see Table 2). 

In situations such as restenosis, and other pathological reactions secondary to interventions, characterized by high levels of IL-6 and other inflammatory cytokines, augmented levels of PRLMs could limit harmful arterial reactions and restenosis. An imbalance between proinflammatory and resolving factors may decide the evolution of atherosclerotic disease [3]. 

There is also favorable data regarding the antithrombotic activity of Maresin 1. In fact, it promotes the hemostatic function of human platelets, but suppresses the inflammatory activity, suggesting its important role in the resolution of thrombotic events [184]. At sites of thrombosis or injury, platelet–neutrophil interaction gives rise to the biosynthesis of PRLMs such as MaR1 which induce resolution [185].

The activity of RvS in thrombosis is directed towards reducing platelet activation [180,186]. RvDs and Lxs also activate macrophages for the phagocytosis of the clot and apoptotic cells [187]. In addition, myocardial ischemia/reperfusion studies in the mouse animal model have established a protective effect of PRLMs with respect to tissue damage and the reduction of ROS [57,58,59] (see Table 1). 

Although until recently lipid mediators have been overlooked as resolvers of inflammation, they are now being reevaluated because of their activities and biosynthesis in the vascular system. In fact, in vitro, RvD1, RvD2, MaR1, LXA4, and protectin D1 reduce the production of inflammatory cytokines, adhesion molecules, and the adhesive effect of leukocytes to endothelial cells during an inflammatory event [6,188,189,190]. These resolving effects result from the ability of these molecules to reduce NF-kB, NO, prostacyclin synthesis, and ROS production [191,192].

Reduced monocyte adhesion to VSMCs and the reduced expression of adhesion molecules have been also observed in vitro studies with inflammatory cytokines after treatment with RvD1, RvD2, and MaR1. Another resolving effect by LXA4, RvD1, RvD2, and RvE1 is to attenuate VSMC migration [193,194].

### 4.3. Protective Function of ω-3 Fatty Acid Diet in Atherosclerotic Progression

Individuals with a high consumption of fish oils show a greater proportion of EPA and DHA-containing phospholipids in particular cell types, compared to individuals consuming plant oil supplements [195]. EPA provides precursors for the production of the anti-inflammatory mediator family of series 3 prostanoids. The synthesis of prostanoids is catalyzed by cyclooxygenase enzymes. The cleavage of EPA by phospholipase A 2 allows EPA to be converted to anti-inflammatory mediators [196].

Human studies have shown the limited endogenous synthesis of docosahexaenoic acid (DHA) from ω-3 fatty acid precursor α-linolenic acid (ALA), therefore supporting the necessity of a diet containing ω -3 PUFAs for optimal health and development [197,198]. Circulating levels of the RvD1 or its precursor DHA are reduced in individuals with symptomatic carotid atherosclerosis [199]. 

Adults receiving EPA+DHA therapy demonstrated significantly greater reductions in circulating levels of proinflammatory cytokines, compared with those receiving placebo therapy [200]. EPA+DHA therapy may be an effective low-risk dietary intervention for assuaging the harmful effects of inflammation.

A large prospective randomized clinical trial of 11,324 patients with a recent myocardial infarction showed a relatively reduced risk of death in response to long chain polyunsaturated fatty acid dietary supplementation [201] (see Table 3).

In another clinical trial, 0.8–1.2 g omega-3 FA supplementation has shown a positive effect in reducing the incidence of major adverse cardiovascular events (MACE), cardiovascular death, and myocardial infarction [202]. A lower rate of myocardial infarction was also observed in people consuming an Inuit diet, characterized by the high consumption of marine mammals and fish [207]. Meta-analyses in adults have shown that the Mediterranean diet, enriched in ω-3 PUFA and oleic acid, exerts a protective effect against cardiovascular diseases [208,209,210,211]. This beneficial effect could be due to their anti-arrhythmic, anti-inflammatory, anti-thrombotic, and hypolipidemic effects, improving vascular function [212,213].

PUFA administration seems to be also protective against aneurysm, a condition characterized by chronic inflammation of the arterial wall that gradually loses its structural integrity by an incessant protease activity. In fact, in animal studies, PUFAs reduced aortic wall inflammation and impairment of the matrix [184]. A recently published trial by Bhatt et al., involving 8179 participants and a 5 year follow-up, reported that patients who received 4 g/d ω-3 PUFA showed a 30% reduction in cardiovascular events, compared with the control group [203]. The JELIS trial, conducted in Japan, recruited 18,645 patients with a total cholesterol of 6.5 mmol/L or greater, and demonstrated a 19% relative reduction in major coronary events after a 1800 mg dose of EPA daily [204].

However, some conflicting data exists [214,215,216]. Other clinical trials on PUFA administration have reported contradictory findings with regard to cardiovascular death and/or major cardiovascular events [205,214,217,218]. Recent studies by Xie et al. suggest that the population most at risk of suffering major cardiovascular events and deaths could benefit more by taking a higher dose of *ω*-3PUFA for a more prolonged intervention period (daily dose × intervention period > 8 g/day). Earlier treatment schedules of only 1 g/d for 1-year ω-3 PUFA supplementation showed no benefits regarding sudden cardiac death, and in fact revealed a small increase in major cardiovascular and cerebrovascular events [206]. Furthermore, a large-scale randomized controlled trial of the effects of marine *ω*-3 fatty acid supplementation on cardiovascular outcomes has reported increased risks of atrial fibrillation (AF), in particular in trials testing >1 g/d. [205]. Recent meta-analyses reflected these discrepancies [219,220].

Therefore, the potential reasons for disparate findings may be dose-related. In addition, isomeric forms of the mediators may show different capabilities or conflicting biological effects. More advanced age or smoking habits may also affect the biosynthesis of lipid mediators [221].

In conclusion, further studies on pro-resolving lipid-mediator biosynthesis and pathways and on diets, with related doses and periods, are needed to understand the best formulations for an effective resolution in relation to cardiovascular disease. Understanding these variables will be necessary for the development of effective drug therapies.

## Figures and Tables

**Figure 1 ijms-24-01637-f001:**
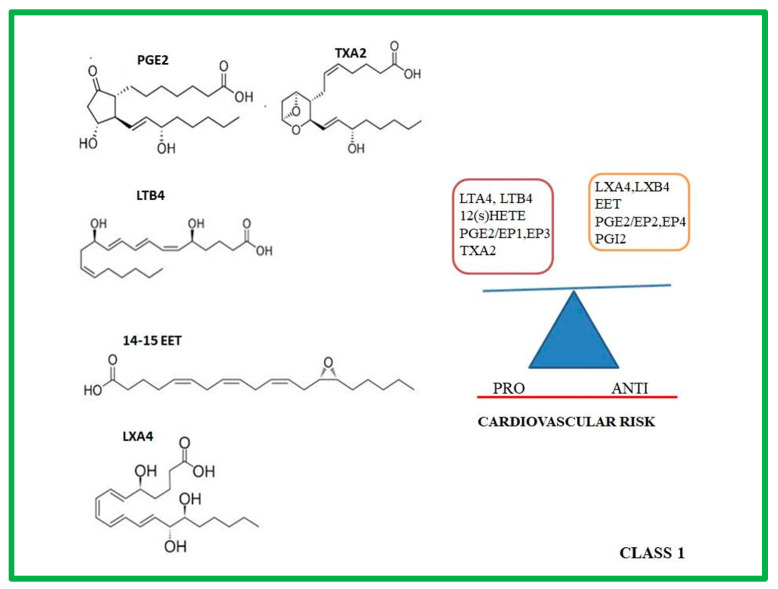
Molecular structure of the principal class 1 lipid mediators (on the left). On the right, the class 1 lipid mediators are divided according to their anti and pro cardiovascular risk properties.PGE2: prostaglandin E2; TXA2: thromboxane A2; 14-15 EET: 14-15 epoxyeicosatrienoic acid; LTB4: Leukotriene B4; LXA4: Lipoxin A4; 12(S)-HETE:12-hydroxyeicosatetraenoic acid; EP1,2,3, 4: prostaglandin Receptors.

**Figure 2 ijms-24-01637-f002:**
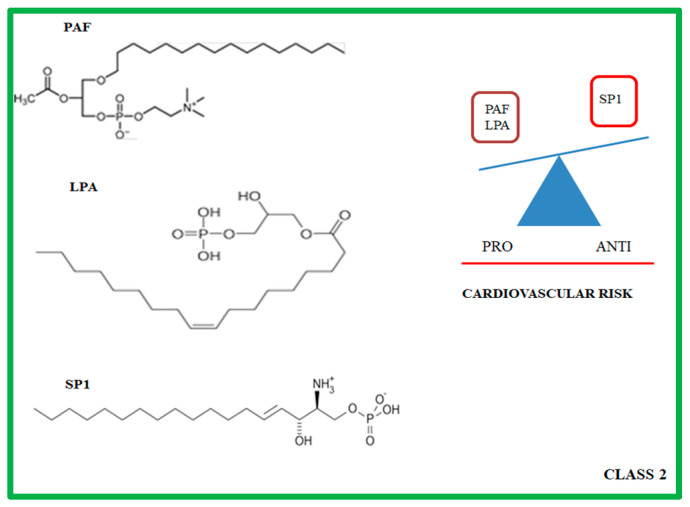
Molecular structure of the principal class 2 lipid mediators (on the **left**). On the **right,** the class 2 lipid mediators are divided according to their anti- and pro-cardiovascular risk properties. PAF: Platelet activating factor; LPA: Lysophosphatidic acid; SP1: sphingosine-1-phosphate.

**Figure 3 ijms-24-01637-f003:**
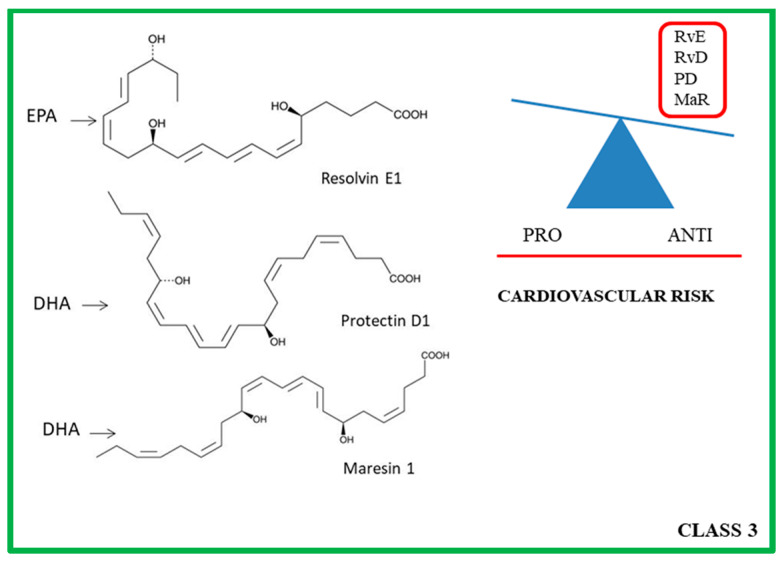
Molecular structure of the principal class 3 lipid mediators (on the **left**). On the **right**, the class 3 lipid mediators are divided according to their anti- and pro-cardiovascular risk properties. RvE: Resolvin E; RvD: Resolvin D; PD: Protectin; Maresin: MaR.

**Figure 4 ijms-24-01637-f004:**
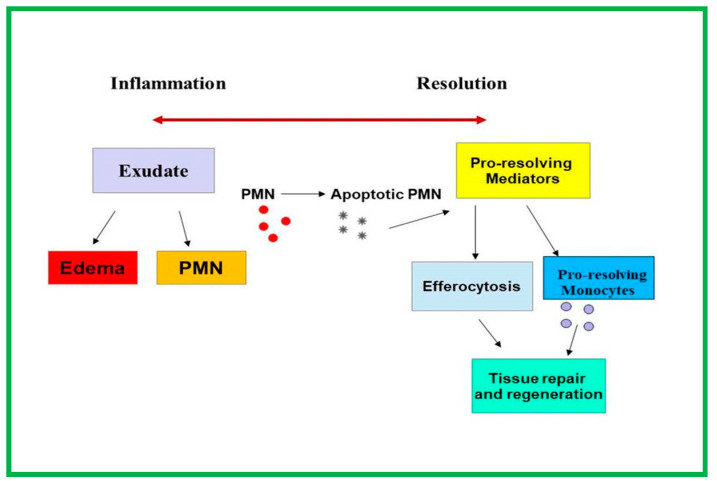
Simplified scheme of biochemical transition from inflammation to resolution. Pro-resolving lipid mediators enhance the clearance (efferocytosis) of apoptotic PNM by macrophages, blocking leukocyte recruitment and favoring tissue repair. PNM (circle in red colour): polymorphonuclear neutrophils; PNM (in gray colour):apoptotic polymorphonuclear neutrophils; stars in blue color: pro-resolving monocytes.

**Table 1 ijms-24-01637-t001:** Studies of Class 1,2,3 lipid mediators in animal models.

Studies in Animal Models	Lipid Mediators	Effects	Ref.
ApoE-deficient mice	↑ TXA2	↑ Atherogenesis	[48]
	↓ PGI2	↑ Atherogenesis	[48]
Mice deficient in LDL receptor/mPGEs 1	↑ PGI2	↓ Atherosclerosis	[49,50]
Mice COX-2 knockout		↑ HBP, Trombosis	[51]
ALOX15-deficient mice	↓ HETES	↑ resistance L-NAME-induced hypertension	[52]
Mice Ang II-induced hypertension	↓EETS↑ sEH	↑ AngII-induced cardiac hypertrophy	[53]
Administration of PAF receptor antagonist (SDZ 63.675) before reperfusion of the ischemic-isolated rabbit heart	↓ PAF	↓ Myocardial injury during reperfusion	[54]
Male LDLr−/− mice (two weeks cholesterol + cacao butter diet) before surgery of carotid artery	↑ LPA in atherosclerotic tissue	Atherosclerotic Lesion Progression	[55]
WT mice administered with soluble carrier for S1P (ApoM-Fc) after I/R injury in heart	↑ SP1 Receptor	↓ Myocardial damage after I/R injury	[56]
Myocardial I/R in mouse	↑ RvE, RvD, PD, MaR	↓ ROS, Inflammation	[57,58,59]

**Table 2 ijms-24-01637-t002:** Studies of Class 1,2,3 lipid mediators in humans.

Lipid Mediators	Effects	Ref
↑ TXA2	Systemic inflammation, myocardial ischemia	[61]
↑ LTs	Inflammation, CV risk, MI	[64,65]
↑ 12(S)-HETE	VSMC relaxation	[66]
↓ EETs	Obstructed CAD	[67]
↑ PAF	Ischemic events	[68,69,70]
↑ LPA	Acute heart attacks and coronary syndrome	[71,72,73]
↓ RvE1	Vascular inflammation	[74,75]
↓ 15-epi-LXA 4	Disease severity	[74,76]

**Table 3 ijms-24-01637-t003:** Trials on ω-3 fatty acid diet in humans.

Study	Design	Supplementation	End Point	Ref.
GISSI-Prevenzione Trials	Clinical trial of 11,324 post-MI patients	Fish oil 850–882 mg EPA and DHA	Reduction Death Risk	[201]
Meta analysis of 14 clinical Trials	135,291 subjects	0.8-1.2 g omega-3 FA(EPA, DHA)	MACE, CVD, and MI	[202]
Trial	8179 participants and 5 years follow-up	4 g/d ω-3 PUFA	30% reduction CVE/ control	[203]
JELIS trial in Japan	18,645 patients with a total cholesterol of 6.5 mmol/L	1800 mg of EPA daily	19% reduction of coronary events	[204]
Trial	Patients suffering of cardiovascular diseases	>1 g/d marine ɷ-3 fatty	Increased AF Risk	[205]
Trial	Partecipants suffering major CVE and death	PUFA daily dose × period > 8 grams/day × years	More benefit/1 g/d for 1-year ω-3 PUFA	[206]

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
