# Peer review of "Classes of Lipid Mediators and Their Effects on Vascular Inflammation in Atherosclerosis"

_ijms, 2023, doi:10.3390/ijms24021637_

Round 1

Reviewer 1 Report

Major change: Organization.

1. Confusing focus of the review. The title of the study showed you want to discuss about the effect of classes of lipid mediators on cardiovascular diseases, which include coronary, cerebrovascular, peripheral arterial, rheumatic, congenital heart diseases, and deep vein thrombosis and pulmonary embolism. In the introduction you said the focus was on atherosclerosis, whereas in the main text, the focus was on their effects on inflammation, which is not specific to the cardiovascular system. I would recommend a re-organization of the review--either focus on atherosclerotic CVD, or just one or two specific types of CVD, so you can discuss into more details. 

2. No clear description of CVD. There is a lack of basics on CVD pathogenesis and lipid metabolism physiology in the cardiovascular system. 

3. Mixture of human and animal studies. It is better to organize all the animal studies in one table and human studies in another one, including some more details, such as the animal models/CVD type of the patient, treatment path and duration, and main findings. 

4. The last section discussing about the PUFA diet on CVD is confusing. If your aim was to determine the effect on CVD from different classes of lipid mediators, why now discussing them as in a diet where you cannot distinguish the types? If your aim was to discuss in general the effect of lipid mediators on CVD, why discuss them as individuals before? The inclusion of this section is fine as it illustrates the application of lipid mediators in real life, but it does not match your scope of this review at its current format. And again, for this section, it is better to make a table showing the basic information regarding the human trials and its potential mechanisms associated/proposed.

Minor change: Formatting. The resolution of the compound structure figures is low and varies among figures; the format of the structures are also different (e.g. the illustration of -COOH). Also, formatting of paragraphs describing Class1, 2, and 3 are different (e.g. class 1, each subclass are under 2.1, 2.2, etc., but classes 2 and 3 are directly described in sub-paragraphs). 

Author Response

Reviewer 1

1.Confusing focus of the review. The title of the study showed you want to discuss about the effect of classes of lipid mediators on cardiovascular diseases, which include coronary, cerebrovascular, peripheral arterial, rheumatic, congenital heart diseases, and deep vein thrombosis and pulmonary embolism. In the introduction you said the focus was on atherosclerosis, whereas in the main text, the focus was on their effects on inflammation, which is not specific to the cardiovascular system.

I would recommend a re-organization of the review--either focus on atherosclerotic CVD, or just one or two specific types of CVD, so you can discuss into more details. 

Thank you very much for the revision.

We took the reviewer's suggestions and tried to make the focus on atherosclerotic CVD by changing both the title and same part of the introduction.

  1. No clear description of CVD. There is a lack of basics on CVD pathogenesis and lipid metabolism physiology in the cardiovascular system. 

We added a more clear description of on CVD pathogenesis and lipid metabolism physiology in the cardiovascular system

  1. Mixture of human and animal studies. It is better to organize all the animal studies in one table and human studies in another one, including some more details, such as the animal models/CVD type of the patient, treatment path and duration, and main findings. 

As suggested by the review we added a table with animal studies and another for human studies

  1. The last section discussing about the PUFA diet on CVD is confusing. If your aim was to determine the effect on CVD from different classes of lipid mediators, why now discussing them as in a diet where you cannot distinguish the types? If your aim was to discuss in general the effect of lipid mediators on CVD, why discuss them as individuals before? The inclusion of this section is fine as it illustrates the application of lipid mediators in real life, but it does not match your scope of this review at its current format. And again, for this section, it is better to make a table showing the basic information regarding the human trials and its potential mechanisms associated/proposed.

We decided to discuss ω -3 PUFA diet in their appropriate section. We added also a table showing the basic information regarding the human trials and its potential mechanisms associated/proposed

Minor change: Formatting. The resolution of the compound structure figures is low and varies among figures; the format of the structures are also different (e.g. the illustration of -COOH). Also, formatting of paragraphs describing Class1, 2, and 3 are different (e.g. class 1, each subclass are under 2.1, 2.2, etc., but classes 2 and 3 are directly described in sub-paragraphs). 

The figures have been remade, also improving the resolution and classes 1, 2, 3 have been described in paragraphs 2, 3, 4 respectively.

Reviewer 2 Report

I reviewed the manuscript “Classes of lipid mediators and their effects on cardiovascular disease“ by Lubrano and colleagues where the authors describe the lipid mediators involved in the regulation of inflammation and resolution and their roles in atherosclerosis. While the topic is of interest, I found the manuscript contains many inappropriate/misleading concepts, errors, and inaccuracies.

I am listing only few of them

1.       The manuscript needs some editing and checking of the English

2.       The abbreviation used in the manuscript not consistent: lipoxin A4 should be abbreviated as LXA4

3.       Figure 1 contains several errors:

a.       The caption above the first structure on the left suggests that the structure can refer to either PGE2, D2, F2a, or PGI2. This is clearly not the reality since each prostaglandin has a distinct chemical structure. The authors should show structure of all the listed PG.

b.       Same consideration applies to the so called “LT” (in this case LTB4)

c.       The concept that TXA2 is unstable is somehow confusing: the majority (if not all) of the chemical mediators (in particular lipid mediators) have a short half life

d.       Although 15-epi-LXA4 was originally identified as a product of aspirin-acetylated COX-2, it is now evident that other endogenous or exogenous acetylating agents can trigger their formation. This should be clarified to avoid confusion.

Author Response

Reviewer 2

I reviewed the manuscript “Classes of lipid mediators and their effects on cardiovascular disease“ by Lubrano and colleagues where the authors describe the lipid mediators involved in the regulation of inflammation and resolution and their roles in atherosclerosis. While the topic is of interest, I found the manuscript contains many inappropriate/misleading concepts, errors, and inaccuracies.

I am listing only few of them

1.       The manuscript needs some editing and checking of the English

2.       The abbreviation used in the manuscript not consistent: lipoxin A4 should be abbreviated as LXA4 

3.       Figure 1 contains several errors:

a.       The caption above the first structure on the left suggests that the structure can refer to either PGE2, D2, F2a, or PGI2. This is clearly not the reality since each prostaglandin has a distinct chemical structure. The authors should show structure of all the listed PG.

b.       Same consideration applies to the so called “LT” (in this case LTB4)

c.       The concept that TXA2 is unstable is somehow confusing: the majority (if not all) of the chemical mediators (in particular lipid mediators) have a short half life

d.       Although 15-epi-LXA4 was originally identified as a product of aspirin-acetylated COX-2, it is now evident that other endogenous or exogenous acetylating agents can trigger their formation. This should be clarified to avoid confusion.

Reviewer 3 Report

An interesting review to describe the lipid mediators of inflammation and resolution and their biochemical activity. Some points should be noted as below.

1) In the Introduction section, the authors wrote “Atherosclerosis begins with the retention of apolipoprotein B-containing lipoproteins within the subendothelial intima of arteries, which leads to the activation of endothelial cells, recruitment of leukocytes, accumulation of cells, extracellular matrix and lipids”. Suggestion: Ecology means that “the interrelations of all organisms which live in one and the same place, their adaptations to their environment, their transformation through the struggle for existence (Haeckel 1868). An recent study propose a model about medical ecology tree of human diseases, and the authors declare that the occurrence, development and outcome of human diseases including belongs to be a spatiotemporal ecological process cardiovascular diseases such as atherosclerosis (AS), that parenchymal cells adaptively interplay with their particular stromal environment such as immune cells and other stromal cells in the complex community (Luo W. Nasopharyngeal Carcinoma Ecology Theory: Cancer as Multidimensional Spatiotemporal Unity of Ecology and Evolution Pathological Ecosystem. Preprints. 2022; 2022100226. Please check, https://www.preprints.org/manuscript/202210.0226/v2). It might be more interesting to talk about these views in the "Introduction section" to make it updated.

2) The concise graphic about molecular and biochemical characteristicrole of â‘  eicosanoids, â‘¡ lysophospholipids, and â‘¢ ω -3 PUFA derivatives  in cardiovascular disease is suggested to be drawn. (Suggestion: the Figure 1/2 in the published paper (the Life Sci. 2022 Dec 1;310:121122) should be consulted).

3) Any interplay roles and sharing common molecules between Prostanoids, LTs , HETEs, LXs, EETs in cardiovascular disease? If it has, please discuss.

4) Likewise, any interaction and molecular characteristic of PAF, LPA and Sp1 in cardiovascular disease? If it has, please discuss something.

5) Similar questions was also raised in “4.1 Molecular and biochemical characteristic of Rvs, Ps and MaRs” section.

Author Response

An interesting review to describe the lipid mediators of inflammation and resolution and their biochemical activity. Some points should be noted as below.

1) In the Introduction section, the authors wrote “Atherosclerosis begins with the retention of apolipoprotein B-containing lipoproteins within the subendothelial intima of arteries, which leads to the activation of endothelial cells, recruitment of leukocytes, accumulation of cells, extracellular matrix and lipids”.

Suggestion: “Ecology” means that “the interrelations of all organisms which live in one and the same place, their adaptations to their environment, their transformation through the struggle for existence” (Haeckel 1868).

 An recent study propose a model about medical ecology tree of human diseases, and the authors declare that the occurrence, development and outcome of human diseases including belongs to be a spatiotemporal ecological process cardiovascular diseases such as atherosclerosis (AS), that parenchymal cells adaptively interplay with their particular stromal environment such as immune cells and other stromal cells in the complex community (Luo W. Nasopharyngeal Carcinoma Ecology Theory: Cancer as Multidimensional Spatiotemporal “Unity of Ecology and Evolution” Pathological Ecosystem. Preprints. 2022; 2022100226. Please check, https://www.preprints.org/manuscript/202210.0226/v2). It might be more interesting to talk about these views in the "Introduction section" to make it updated.

2) The concise graphic about molecular and biochemical characteristicrole of ? eicosanoids, ? lysophospholipids, and ? ω -3 PUFA derivatives  in cardiovascular disease is suggested to be drawn. (Suggestion: the Figure 1/2 in the published paper (the Life Sci. 2022 Dec 1;310:121122) should be consulted).

3) Any interplay roles and sharing common molecules between Prostanoids, LTs , HETEs, LXs, EETs in cardiovascular disease? If it has, please discuss.

4) Likewise, any interaction and molecular characteristic of PAF, LPA and Sp1 in cardiovascular disease? If it has, please discuss something.

5) Similar questions was also raised in “4.1 Molecular and biochemical characteristic of Rvs, Ps and MaRs” section.

Round 2

Reviewer 1 Report

The authors stressed the corresponded comments and the changes added to the logic flow and readability of the manuscript. Except for some minor changes in format (e.g. 2.4 xxx: VS. 3.4 xxx;), it is good for publish.

Author Response

We thank the Reviewer 1 for the positive judgment regarding the manuscript after the modifications made.

Reviewer 2 Report

The authors were highly willing to address my previous questions and they effort is acknowledged.

However, there are still some errors to fix in my opinion

1.     Figure 1 shows the structure of PGE2 and LTB4. Please indicate it in the figure and legend to avoid confusion

2.     Resolvin E1 stereochemistry is wrong: please correct with the exact configuration of C12 from S into R and the C6 double bond geometry from E to Z. For your reference, here is the full IUPAC name and stereochemistry of RvE1: 5S,12R,18R-trihydroxy-6Z,8E,10E,14Z,16E-eicosapentaenoic acid

3.     The structure of Mar1-n3DPA is also wrong

Author Response

We Thank the Reviewer 2 for the positive comment and corrected the errors indicated. 

 1) we reported PGE2 and LTB4 in the Figure 1 and in the legend.

 2) the figure 3 was changed reporting the correct structures of Resolvin E1 and Maresin 1 by “Cayman Chemical”.

Reviewer 3 Report

The author's revision made nearly no changes, it was not qualified.

Author Response

We are not agree with the opinion of the Reviewer because we think that the manuscript has been improved. Moreover a high English revision has been done by a native English graduated

Round 3

Reviewer 3 Report

No other Q.